# OpenReview forum: "Energy-Driven Steering: Reducing False Refusals in Large Language Models"
_ICLR.cc/2026/Conference — ICLR 2026 Conference Withdrawn Submission_

### Official Review · Reviewer_iri4 · 2025-10-29

**Soundness:** 2
**Presentation:** 3
**Contribution:** 2
**Rating:** 2
**Confidence:** 4

**Summary:**

This paper proposes Energy-Driven Steering, which trains energy-based models to separate desired and undesired model behaviour with contrastive learning. The experimental results show improvement in the separation of the refusal and over-refusal representation, while maintaining the general capability.

**Strengths:**

The Energy-based model is well motivated and works well to separate desired and undesired behaviours.
The author also extend the evaluation to multi-turn jaibreak datasets, which further test the robustness of the method.

**Weaknesses:**

The main weakness of the paper is the lack of text generation quality evaluation.
There is no quantitative or qualitative evaluation of the model generation, which raises the concern of the text generation quality.

Such an evaluation is especially important as the intervention is quite strong based on the method. There is also no mechanism in the method to ensure the text generation quality, which requires such an evaluation. It is very likely the gradient descent intervention will interfere the text quality.

**Questions:**

It is not very clear in the method how the intervention is applied token-wise.
How exactly is the intervention done? Do you do intervention on all the response tokens at once? If so,  how can we make sure the new tokens still construct meaningful sentences if those tokens are not generated auto-regressively?

Are there any specific reasons why the capability task performance is kept? Why does simply doing gradient descent on the energy landscape will retain such capability?

---

### Official Review · Reviewer_ksYb · 2025-10-31

**Soundness:** 2
**Presentation:** 2
**Contribution:** 3
**Rating:** 4
**Confidence:** 3

**Summary:**

The paper presents an new method for steering safety refusal in LLMs. The key idea is to use a lightweight, externally trained Energy-based Model (EBM) to steer internal activations at inference time. This contrasts to existing approaches which learn e.g. a single steering vector that is static. The paper tests the capabilities of Llama and Qwen models on datasets that cover safety, false refusal, general capabilities and multi-turn attacks.

**Strengths:**

- Method. The paper presents a novel, lightweight method for model steering. The results show that the idea has merit, improving a range of prior methods.

- Evaluation. The paper tests the approach on a wide array of benchmarks including safety, false refusal and general capabilities, similar to [Wang et al., 2024](https://arxiv.org/pdf/2410.03415)

**Weaknesses:**

- Presentation. While overall written clearly, I felt the paper could do a better job in the details. For example, methods are not linked to their actual papers, not explicitely. I find it confusing to understand what is meant by which method in the paper. Concrete example:  ", unlike methods such as Surgical Vector and AdaSteer, which show a degradation in safety performance (i.e., higher compliance with harmful requests)". What method is referred to with "Surgical Vector"? If it is Wang et al., (2024), line 137 introduces it as "VA (Vector Ablation) (Wang et al., 2024)", but the naming does not return in the experimental result discussion.

- The main point of the paper is the introduction of the EBM model, with the motivation to obtain a dynamic steering method, i.e. steering that is not static but adapts on an instance level. However, this key selling point is not evaluated - as results are given as aggregate over benchmarks. So it remains unclear whether the contribution is from the dynamic aspect, or just by using a different way to obtain the steering (an external model). I think the paper would be stronger by including an experiment that shows more directly the differences between a static steering vector and the dynamic steering the paper suggests.

**Questions:**

- Prior work has introduced partial orthogonalization as a way to obtain a more dynamic way of steering (Wang et al., 2024). Have you tested that? Do you see that as alternative to your steering coefficienct term? If different, how does it differ?

- I am confused of the naming of methods in your paper / results. Could you provide a clearer mapping of prior work methods and their corresponding references?

- The efficiency analysis: I do not understand why EBM steering would be more efficient than surgical vector steering. EBM uses a call to an external model, while surgical vector steering a single vector ablation. Did you average over different runs? I wonder how much variation you would get in your inference time estimates by including several runs and standard deviation over them. THen the differences would probably be negligible to some methods.

---

### Official Review · Reviewer_kySW · 2025-11-02

**Soundness:** 3
**Presentation:** 3
**Contribution:** 2
**Rating:** 4
**Confidence:** 4

**Summary:**

The paper proposes Energy-Driven Steering (EDS), a fine-tuning-free method to reduce false refusals at inference time. A lightweight external energy-based model (EBM) is trained to assign low “energy” to desirable hidden states (helpful answers to benign prompts; safe refusals to harmful prompts) and high energy to undesirable ones (false refusals; jailbreaks). During generation, gradients of this energy are used to nudge hidden states toward low-energy regions, aiming to improve benign compliance without sacrificing safety or general capabilities. Experiments across several open models (Llama-3.1-8B, Llama-2-7B, Qwen3 family) report large gains on over-refusal benchmarks with near-baseline safety, plus multi-turn robustness and modest latency overhead.

**Strengths:**

(1) Clear, modular idea: Decoupling behavior steering from the base LM via an external EBM is conceptually clean and deployment-friendly (no weight updates)

(2) Empirical breadth: Evaluates across multiple families/sizes and reports three axes: false-refusal, safety, and general capabilities; also includes multi-turn robustness and efficiency (latency) analyses.

(3) Theory support: Provides a principled view of steering as energy minimization and ablations on hyperparameters.

(4) Low overhead: Inference-time cost appears small relative to other steering baselines.

**Weaknesses:**

(1) Novelty relative to activation steering and single-vector methods: The paper’s mechanism (learning a classifier/energy over hidden states and applying gradient-based perturbations) is close in spirit to prior activation steering / vector ablation approaches. The related-work section acknowledges activation steering and Surgical vector / single-vector ablation [1]. However, the novelty gap feels narrower than claimed: EDS can be interpreted as learning a more expressive steering field, with a heuristic labeler to form positive/negative hidden-state sets. I recommend the authors more explicitly contrast EDS against “Surgical, Cheap, and Flexible...” both conceptually and empirically (e.g., where exactly EDS helps over learned per-layer multi-direction steering, and on which inputs).

(2) Baseline coverage for simple SFT approaches to over-refusal: Recent work (e.g., datasets like FalseReject [2]) shows that targeted supervised fine-tuning can reduce over-refusal while preserving safety to a very high extent. The current paper includes RLHF-style baselines (e.g., Defender/Self-Play + SFT) and several steering baselines, but does not include a simple SFT-only baseline trained specifically on over-refusal data. Given the engineering cost of running an auxiliary EBM online at every step, a strong SFT baseline on over-refusal data should be included for fairness and practicality.

(3) Generalization of the learned shift across datasets/domains: Training data for the EBM is drawn from SafeMedEval-21K (medical-domain prompts) while evaluation spans mixed-domain safety and false-refusal suites. That suggests some cross-domain generalization of the energy field, which is encouraging—but the paper doesn’t present a train/eval protocol to isolate generalization (e.g., leave-one-benchmark-out or leave-one-category-out). A focused study showing EBM trained on dataset A and tested on disjoint dataset B (and vice-versa) would strengthen claims of robustness. Please clarify whether any such disjointness checks were performed.

[1] Wang, Xinpeng, et al. "Surgical, cheap, and flexible: Mitigating false refusal in language models via single vector ablation." arXiv preprint arXiv:2410.03415 (2024).
[2] Zhang, Zhehao, et al. "Falsereject: A resource for improving contextual safety and mitigating over-refusals in llms via structured reasoning." arXiv preprint arXiv:2505.08054 (2025).

**Questions:**

(1) Novelty relative to activation steering and single-vector methods:
Can the authors clearly articulate what conceptual or empirical advantage Energy-Driven Steering offers over existing activation-steering or single-vector ablation methods (e.g., Surgical, Cheap, and Flexible)? In particular, on which input types or safety–helpfulness trade-offs does EDS show distinct improvements?

(2) Baseline coverage for simple SFT approaches:
Could the authors add or discuss results for a straightforward supervised fine-tuning baseline trained specifically on over-refusal data (e.g., FalseReject-style) to quantify the added benefit of the EBM-based steering compared with this much simpler alternative?

(3) Generalization across datasets/domains:
Was any experiment performed where the EBM was trained on one dataset and evaluated on a disjoint or unseen benchmark to verify cross-domain generalization? If not, could the authors comment on how well they expect EDS to transfer across domains and what limits that transfer?

---

### Note · Authors · 2025-11-23

**Comment:**

Dear AC, SAC, PC, and Reviewers,

Thank you all for taking the time to carefully review our manuscript. We have thoroughly read your valuable comments and feedback. After careful consideration, we have decided to withdraw this submission to allow sufficient time for substantial revisions. We sincerely appreciate your insightful suggestions, which will greatly help us improve the work. We look forward to resubmitting a significantly improved version in the future.

Thank you again for your time and support.

Best regards,

Authors

**Withdrawal Confirmation:**

I have read and agree with the venue's withdrawal policy on behalf of myself and my co-authors.